Oleanolic acid and moderate drinking increase the pancreatic GLP-1R expression of the β-cell mass deficiency induced hyperglycemia

Xu Li maolixu@hotmail.com 1
Hu Ruibin 1
Jois Shreyas Venkataraman 2
Zhang Lei 502266827@qq.com 3
1 Medical experiment center, Shaanxi University of Chinese Medicine , Xianyang , China
2 Health Sciences Building, University of Saskatchewan , Saskatoon , Canada
3 Shaanxi Eye Hospital, Xi’an People’s Hospital , Xi’an , China
Amdare Dr. Nitin
Electronic publication date: 2023 Jul 24
Publication date: 2023
Volume: 11
Electronic Location ID: e15705
Received 2023 Feb 16; Accepted 2023 Jun 15
Copyright: ©2023 Xu et al.
Copyright year: 2023
Copyright holder: Xu et al.
License: This is an open access article distributed under the terms of the Creative Commons Attribution License, which permits unrestricted use, distribution, reproduction and adaptation in any medium and for any purpose provided that it is properly attributed. For attribution, the original author(s), title, publication source (PeerJ) and either DOI or URL of the article must be cited.
License URL: https://creativecommons.org/licenses/by/4.0/

Keywords: Diabetes, β-cell, α-cell, GLP-1R, Oleanolic acid, Moderate drinking

Funding: The authors received no funding for this work.

==============================
Background

Oleanolic acid (OA) and moderate drinking have been reported to attenuate diabetes. However, the underlying mechanism of OA and moderate drinking alone or in combination on the islet β-cell deficiency induced diabetes is not fully elucidated.

Methods

Male Sprague Dawley (SD) rats were intraperitoneally injected with 55 mg/kg streptozotocin (STZ) to induce β-cell deficiency. OA, 5% ethanol (EtOH), or a mixture of OA in 5% ethanol (OA+EtOH) were applied to three treatment groups of hyperglycemia rats for 6 weeks.

Results

STZ caused the increase of fast blood glucose (FBG) level.OA and EtOH treatment alone or in combination decreased the STZ increased FBG level during the 6 weeks of treatment. In addition, OA treatment also significantly increased the β-cell to total islet cell ratio. Both EtOH and OA+EtOH treatments promoted the increase of total islet cell number and α-cell to β-cell ratio when compared to OA group. STZ induced hyperglycemia dramatically reduced the glucagon-like peptide-1 receptor (GLP-1R) positive cells in islets, all the three treatments significantly increased the pancreatic GLP-1R positive cell number. In the meantime, STZ induced hyperglycemia suppressed the insulin mRNA expression and boosted the glucagon mRNA expression. EtOH and OA+EtOH treatments increased the insulin mRNA expression, but none of the 3 treatments altered the elevated glucagon level.

Conclusion

GLP-1R positive cell ratio in islets is crucial for the blood glucose level of diabetes. OA and 5% ethanol alone or in combination suppresses the blood glucose level of β-cell deficiency induced diabetes by increasing islet GLP-1R expression.

Introduction

Glucose homeostasis is regulated primarily by β-cell produced insulin and α-cell produced glucagon. Insulin is the only glucose-lowering hormone and its secretion is crucially dependent on pancreatic β-cell function. Lacking of β-cell mass or function lead to hyperglycemia. The β-cell deficiency caused hyperglycemia was commonly found in both type-1 (T1D) and type-2 diabetes (T2D) (Weir & Bonner-Weir, 2013). A possible therapy for diabetes is restoring β-cell function by regenerating pancreatic β-cells (Aguayo-Mazzucato & Bonner-Weir, 2018). Glucagon-like peptide-1 receptor (GLP-1R), mainly found in pancreatic β-cell can also be expressed in α-cells under diabetic conditions (Nakashima et al., 2018). A recent study indicated that GLP-1 improves glucose levels via regulating β-cell function while promoting islet α-cell to β-cell trans-differentiation (Lee et al., 2018). Therefore, GLP-1R presented as a potential target to regulate β-cell function.

Oleanolic acid (OA), a plant derived triterpenoid has a great potential in treating chronic diseases (Ayeleso, Matumba & Mukwevho, 2017). OA can lower postprandial hyperglycemia of STZ induced diabetic rats (Khathi et al., 2013) and enhance glucose homeostasis of pre-diabetic rats (Gamede et al., 2018). OA has shown to enhance the survival of pancreatic islets (Nataraju et al., 2009), it is not clear if OA administration can also influence the islet cell number or cell type during the β-cell deficient state.

Even though there is a drastic debate on whether moderate drinking is beneficial for diabetes, lots of surveys or clinical cohort studies support that moderate alcohol drinking reduces the incidence of T2D (Kerr et al., 2019; Ma et al., 2022; Polsky & Akturk, 2017). For people who have T2D, light to moderate drinking can prevent (Wakabayashi, 2018) or even ameliorate diabetic related complications (Abraham et al., 2016; Fenwick et al., 2015; Golan, Gepner & Shai, 2019). A study showed that up to 63% of T1D patients are reported to use alcohol monthly (Petry et al., 2018). Among T1D patients, mild alcohol drinkers had a higher glucose disposal rate and a lower proportion of nephropathy or retinopathy events (Harjutsalo et al., 2014). Light to moderate alcohol drinking could be beneficial for both T1D and T2D, and the average dosage of moderate drinking varied from 5-40g ethanol/day (Li et al., 2016). Furthermore, whether moderate consumption of low concentration alcohol will be beneficial for diabetes when islet β-cells are diminished remains unknown.

Recently, OA enriched olive oil has been reported to prevent the development of diabetes in pre-diabetic patients (Santos-Lozano et al., 2019) which displayed OA as a promising ingredient to make functional food for diabetic patients. Some studies of fermented alcohol suggest that the effects of metabolism disease are not only due to ethanol but also due to the content of the active compounds in the beverage (Arranz et al., 2012; Cerrillo et al., 2019; Franc, Muselik & Vetchy, 2018). Both moderate drinking and OA had been reported to benefit diabetes, whether the mixture of OA and low concentrated alcohol can still perform anti-diabetes function needs more investigation. In this study, the effects of OA, 5% ethanol, and their mixture on hyperglycemia rats were tested, and the potential mechanism was verified by investigating changes in the islet cell mass and type.

Material and Methods

Animals and experimental procedure

Male Sprague Dawley (SD) rats (7–8 weeks of age) were obtained from the Experimental Animal Center of Xi’an Jiaotong University (SCXK 2012-003). The animals were maintained in a 12 h/12 h light and dark cycle at the temperature of (23 ± 2) °C and free access to food and water. All animal experiments were carried out following the National Institutes of Health guide for the care and use of Laboratory animals and the guidelines of the Experimental Animal Ethics Committee of Shaanxi University of Chinese Medicine (SUCMDL20160113001).

Fifty rats were injected with 55mg/kg STZ to generate hyperglycemia according to our previous method (Xu, Jois & Cui, 2022). Rats with blood glucose levels ≥16.7 mmol/L were selected for further treatment. Hyperglycemia rats were randomly divided into the hyperglycemia group (10 rats), OA group (10 rats), EtOH group (8 rats), and OA+EtOH group (10 rats). 10 rats were injected with an equal dose of 0.1 M sodium citrate buffer (pH = 4.4) as the control group. Control and hyperglycemia groups were treated with normal saline (10ml/kg/d), the OA group was administered with 100 mg/kg/d of OA suspension, the EtOH group was administered with 5% V/V of ethanol (10 ml/kg/d) which was based on previous research (Adaramoye & Oloyede, 2012; Hu et al., 2014), and the OA+EtOH group was treated with 100 mg/kg OA mixed in 5% V/V of ethanol. The body weight and FBG concentrations were measured every week, food intake was recorded regularly. Rats were anesthetized with 2% isoflurane after 6 weeks of treatment, then blood, livers, and pancreas were collected. The pancreas tissue used for qPCR, was immediately snap-frozen in liquid nitrogen and kept in RNase free tubes at −80 °C before RNA extraction. The liver and pancreas used for the sections were firstly rinsed with ice-cold PBS, then fixed with 4% paraformaldehyde for 12 h. After sample collection, all animals were euthanized with 5% of isoflurane.

HE staining

Paraffin embedded pancreatic tissue was sectioned into 3 µm and placed onto positively charged glass slides. After de-waxed and rehydrated sections were stained in hematoxylin solution for 5 min. Followed by eosin red staining for 15 s, samples were dehydrated and sealed with neutral balsam.

Immunohistochemical staining

Pancreatic sections were de-waxed and rehydrated, then incubated with 3% hydrogen peroxide for 20 min. After antigen retrieval, sections were blocked with a 5% BSA for 30 min. Rabbit anti-insulin (1:50; BM4310; Boster, Wuhan, China) or rabbit anti-GLP-1R (1:150; bs-1559R; Bioss, Beijing, China) was applied as the primary antibody to detect insulin and GLP-1R expression. The labeled avidin-biotinylated rabbit IgG kit (SA1022, Boster, Wuhan) was selected to amplify the antibody signal. The antibody-antigen complex was visualized by incubating the samples in DAB chromogen (AR1022, Boster, Wuhan) and counterstained with Harris hematoxylin.

Immunofluorescent staining

Sections were firstly incubated with primary antibodies rabbit anti-insulin (1:50; BM4310; Boster, Wuhan, China) and mouse anti-glucagon (1:200; BM1621; Boster, Wuhan). After thoroughly washing with TBST, samples were then incubated with FITC conjugated goat anti-rabbit IgG (1:50; BA1105; Boster, Wuhan, China) and Cy3 conjugated goat anti-mouse IgG (1:50; BA1031; Boster, Wuhan, China). Finally, sections were cover-slipped with Boster AR1109 (Boster, Wuhan, China) mounting medium.

PAS staining

Liver glycogen was measured using a Glycogen Periodic Acid Schiff(PAS/Hematoxylin) Stain Kit (G1281; Solarbio, Beijing, China) and the assay was performed according to the manufacturer’s instructions.

RNA extraction and qRT-PCR

Total RNA was extracted from pancreas tissue using the Takara RNAiso Plus reagent (D9109, Takara, Japan). The concentration of total RNA was determined by spectrophotometry (Nanodrop 2000; Thermo Scientific, Waltham, MA, USA). Reverse transcription for cDNA was synthesized from total RNA using a PrimeScript RT Master Mix Kit (RR037A; Takara, Shiga, Japan). The real-time PCR was carried out on an Applied Biosystems 7500 Real-Time PCR System (Applied Biosystems, Foster City, CA, USA) with an SYBR Green premix EX Taq II kit (RR820A; Takara). The results for each specific gene were normalized to the ribosome 18s RNA gene. The expression level of each gene was calculated by the 2−ΔΔCt method (Livak & Schmittgen, 2001). The primer sequences used in this study were:18sRNA (F) GAC TCA ACA CGG GAA ACC TCA CC and (R) ACC AGA CAA ATC GCT CCA CCA AC; Ins1 (F) GGA CCC GCA AGT GCC ACA AC and (R) TGA TCC ACA ATG CCA CGC TTC TG; Gcg(F) ACC GTT TAC ATC GTG GCT GGA TTG and (R) TCT GGC GTT CTC CTC CGT GTC.

Image acquisition and statistical analysis

An Olympus BX41 microscope (Olympus, Tokyo, Japan) was used to image HE staining, PAS staining, and Immunohistochemical staining (IHC). Immunofluorescent staining (IFC) pictures were acquired by a Zeiss LSM 5 Pascal confocal microscope (Jena, Germany).

Cell counting and positive staining area selection method was used as our previous work (Xu, Jois & Cui, 2022). Randomly selected images from 3–5 animals of each group were used for analysis with image J software. For PAS staining the glycogen expression level was demonstrated semi-quantitatively as the percentage of the PAS positive staining area/liver area. For IHC and IFC staining, insulin, glucagon, and GLP-1R positive cells were counted from the captured islets. The liver tissue pictures or islet number used for analysis was marked as per figure notes.

All the data were reported as the mean ± SEM. The statistics were processed using the GraphPad Prism 8 software (GraphPad Software Inc., San Diego, CA, USA). If the data passed the normality test or data set ≥30, two-tailed t-test was used for comparison between the two groups, and the multiple groups’ variance analysis was used one-way ANOVA with post hoc Tukey–Kramer test. Nonparametric Mann–Whitney test was performed for data which not passed the normality test. A p value less than 0.05 was considered significant difference.

Results

OA and moderate drinking alone or in combination suppressed STZ induced hyperglycemia

STZ injection significantly increased the FBG level as compared to the control group (p < 0.001); the average FBG level in OA, EtOH, and OA+EtOH groups fluctuated from 11.3–28.8 mmol/L throughout the treatment but were lower than in the hyperglycemia (25–32.1 mmol/L) group (Fig. 1A). When counted the 6 weeks together, the average FBG level of OA, EtOH and OA +EtOH groups were significantly decreased compared to the hyperglycemia group (p < 0.05, Fig. 1B). The body weight in the hyperglycemia group was dramatically decreased compared to the control group, but it did not differ from all the other treatment groups (Fig. 1C). Diabetic rats consumed more food than control, whereas the food intake of OA, EtOH and OA+EtOH were not different from the hyperglycemia group (Fig. 1D).

Figure 1 OA and moderate drinking alone or in combination ameliorate STZ induced hyperglycemia.

(A) FBG level during treatment. (B) Average FBG level of 6 weeks’ treatment. (C) Body weight changes during treatment. (D) Average food intake during treatment. (E) Representative HE staining of the pancreas. * p < 0.05, ** p < 0.01; n = 7 − 10; Scale bar = 50 µm.

HE staining showed that the islets in the control group were intact and in oval shape, but the islets in the hyperglycemia group were destroyed and irregular in shape. All the 3 treatment groups affected the islet size. The islets in the OA group were small and deformed; the islets size of EtOH group was augmented and the islets in OA+EtOH group were irregular in size and tend to be bigger than that in hyperglycemia group (Fig. 1E, dotted circle).

OA enhanced the moderate drinking induced liver glycogenolysis of STZ induced hyperglycemia

During the fasting state, hepatic glycogenolysis is crucial to maintain blood glucose level, so glycogen level can represent the status of liver glycogenolysis. To determine the live glycogen changes, PAS staining was performed. The PAS positive staining in the control group was located in the hepatocellular soma and sparsely distributed, after STZ treatment the PAS positive signal was intensively expressed in the liver (Fig. 2A, star). The PAS staining pattern in the OA group was similar to the hyperglycemia group; PAS positive signal was mainly found around the portal area and central vein of the EtOH group, while very less PAS staining could be observed in the OA+EtOH group (Fig. 2A). The concentration of liver glycogen level was calculated as the percentage of PAS positive area. Rats in the hyperglycemia group had notably elevated glycogen level compared to the control group; OA treatment significantly increased the glycogen level; EtOH treatment down-regulated the heightened glycogen level and OA+EtOH neutralized the hyperglycemia boosted glycogen expression (Fig. 2B).

Figure 2 OA enhanced the moderate drinking induced liver glycogenolysis of STZ induced hyperglycemia.

(A) PAS staining of liver. (B) Percentage of glycogen area in liver. ** p < 0.01, *** p < 0.001; Scale bar = 50 µm; n = 36 − 45; stars point the PAS positive area.

Ethanol neutralized OA raised islet β-cell ratio of STZ induced hyperglycemia

To detect the islet β-cell number, pancreatic tissues were immunohistochemically stained for insulin. As expected, the insulin positive staining was distributed all over the pancreatic islets of the control group, and there were few dark stained β-cells randomly located in the islets’ residue of hyperglycemia group. The insulin expression pattern in the islets of the OA group was similar to the hyperglycemia group, except few insulin positive cells in acinar cells outside of islets. In the EtOH and OA+EtOH group, the islets were larger and the insulin-positive cells were irregularly distributed among the islets (Fig. 3A, red dotted circle).

Figure 3 Ethanol neutralized OA raised islet β-cell ratio of STZ induced hyperglycemia.

(A) IHC of insulin in pancreas. (B) Insulin positive cell number in islet. (C) Total cell number in islet. (D) Insulin positive cell ratio in islet. (E) Average islet area of different groups. * p < 0.05, ** p < 0.01, *** p < 0.001; Scale bar = 50 µm; n = 50 − 75.

The islet β-cell number dramatically decreased after STZ injection and OA, EtOH or OA+EtOH treatment did not significantly affects the β-cell mass of hyperglycemia rats (Fig. 3B). The total islet cell number in hyperglycemia group was significantly decreased compared to the control group. EtOH and OA+EtOH treatments significantly increased total islet cell number compared to OA (Fig. 3C). In the control group, the proportion of β-cell to total islet cell was about 72%. The β-cell ratio for the hyperglycemia group was only 1/3 as in the control group. OA significantly upregulated the β-cell to total islet cell ratio compared to the hyperglycemia and EtOH group (Fig. 3D), while EtOH increased the islet area as compared to the OA and hyperglycemia group (Fig. 3E).

OA and moderate drinking had opposing effects on the α-cell ratio of STZ induced hyperglycemia

The two major islet cell types α-cell and β-cell secrete insulin or glucagon to maintain glucose homeostasis. Double staining of insulin and glucagon was performed to investigate the α-cell and β-cell expression patterns in the islet. In the control group, the β-cells were dominantly expressed in the center of the pancreatic islet which was surrounded by a ring of α-cells. There were only a few β-cells surrounded by the massively expressed α-cells, in the islet of the hyperglycemia group. The general expression pattern of α-cell and β-cell in OA, EtOH, or OA+EtOH group was similar to hyperglycemia group based on the observation of images (Fig. 4A).

Figure 4 OA and ethanol had opposing effects on the α-cell ratio of STZ induced hyperglycemia.

(A) IFC double labeling of insulin and glucagon in pancreas. (B) Insulin positive cell number in islet. (C) Insulin plus glucagon positive cell number. (D) Insulin positive cell ratio. (E) Glucagon positive cell number in islet. (F) Glucagon to insulin positive cell ratio. (G) Glucagon positive cell ratio. * p < 0.05, ** p < 0.01, *** p < 0.001; n.s, not significant different; Scale bar = 50 µm; n = 31 − 72.

STZ induced hyperglycemia decreased the number of β-cell in the islet and OA, EtOH, or OA+EtOH treatment did not significantly alter the β-cell cell expression (Fig. 4B). Interestingly, the combined cell number of α and β-cells in EtOH and OA+EtOH groups were raised when compared to the OA group (Fig. 4C), which could either be a result of the increase in α-cell number (Fig. 4E) or α-cell/ β-cell ratio (Fig. 4F). The β-cell to α-cell plus β-cell ratio was increased in the OA group when compared to hyperglycemia or EtOH group (Fig. 4D) and it was inversely related to α-cell to the β-cell ratio (Fig. 4F) or α-cell to α-cell plus β-cell ratio (Fig. 4G).

OA and moderate drinking increased the GLP-1R positive cell ratio of STZ induced hyperglycemia

GLP-1R in the islet contributes to glucose regulation by improving β-cell function and promoting islet α-cell to β-cell trans-differentiation (Lee et al., 2018). To test the effects of OA, EtOH, or OA+EtOH on pancreatic GLP-1R expression, pancreases were immuno-histochemically stained for GLP-1R. GLP-1R was expressed all over the islet of the control group. In the islet of hyperglycemia group, GLP-1R was stained with brown color and randomly distributed in some cell soma. The expression pattern of GLP-1R in the OA group was similar to hyperglycemia group but stained in a darker color. GLP-1R positive cells were disorderly located in the expanded islet of the EtOH and OA+EtOH group (Fig. 5A).

Figure 5 OA and moderate drinking increased GLP-1R positive cell ratio of STZ induced hyperglycemia.

(A) IHC of GLP-1R in pancreas. (B) GLP-1R positive cell number in islet. (C) Total cell number in islet. (D) GLP-1R positive cell ratio in isle. (E) Average islet area of different groups. * p < 0.05, ** p < 0.01, *** p < 0.001; Scale bar = 50 µm; n = 45 − 54.

In comparison with hyperglycemia group, OA, EtOH or OA+EtOH treatment increased the number of positive GLP-1R cells (Fig. 5B) as well as GLP-1R positive cell to total islet cell ratio (Fig. 5D). Although the total islet cell number of EtOH or OA+EtOH was augmented when compared to the OA group (Fig. 5C), only EtOH had a larger islet size compared to OA and hyperglycemia group (Fig. 5E).

OA and 5% ethanol rescue hyperglycemia suppressed Ins1 mRNA expression

To verify the gene expression pattern of insulin and glucagon, Ins1 and Gcg mRNA expression levels were tested. STZ treatment significantly suppressed the Ins1 expression, and OA, EtOH, or OA +EtOH treatment upregulated the Ins1 expression, but only EtOH and OA +EtOH groups were significantly different from hyperglycemia group (Fig. 6A). Compared to the control, hyperglycemia dramatically increased Gcg expression; the OA group had a lower Gcg level, but EtOH and OA+EtOH groups had higher Gcg levels than hyperglycemia; these groups were not statistically different from each other (Fig. 6B).

Figure 6 OA and moderate drinking rescue hyperglycemia suppressed Ins1 mRNA expression.

(A) Pancreatic Ins1 RNA expression. (B) Pancreatic Gcg RNA expression. * p < 0.05, ** p < 0.01; n = 4.

Discussion

The anti-diabetes effects of OA had been reported by many researchers. OA can significantly enhance insulin secretion from the in-vitro β-cell lines and improve acute glucose-stimulated insulin secretion in isolated rat islets (Teodoro et al., 2008). The use of OA (10–1700 mg/kg) in several studies displayed a low toxicity profile (Silva, Oliveira & Duarte, 2016). When pancreatic β-cells were massively destroyed by STZ in-vivo, OA treatment slightly increased insulin RNA expression (Fig. 6A), and it did not significantly increase the β-cell number (Fig. 3B). However, we still observed that the FBG level of hyperglycemia rats was suppressed by the OA treatment (Fig. 1B). A previous study proposed that OA administration reversed dyslipidemia in T2D mice possibly by the suppression of hepatic glucose production instead of inducing insulin secretion or pancreas morphology change (Zeng et al., 2012). Glycogenesis is crucial for glucose regulation, our result showed that OA down regulated the FBG level at least partially via increasing the liver glycogen level (Fig. 2). On the other hand, OA suppressed the proliferation of α-cells, which causes a decrease in the α-cell to the β-cell ratio (Figs. 4E and 4F), resulting in balanced insulin and glucagon secretion to maintain lower glucose level.

There is a controversy on whether moderate drinking is beneficial for diabetes. Most of the moderate alcohol drinking results were based on surveys or clinical trials, and there was limited animal research about the impact of moderate alcohol drinking on diabetes. In this study, moderate drinking of 5% ethanol showed the potential to suppress the FBG of STZ induced diabetes (Fig. 1B). 5% ethanol significantly reduced the FBG level of diabetic rats in the first 2 weeks, but this effect was not significant from the third week (Fig. 1A). Long term research suggested that moderate drinking did not affect the glycemic level of diabetic patients (Hirst et al., 2017), but improved insulin sensitivity (Zilkens & Puddey, 2003). Our results indicate that 5% ethanol has instant effects on FBG levels in newly generated diabetic rats, and the long-term effects on diabetes need more evidence. Even though moderate drinking showed the ability to down regulate hyperglycemia, there is a potential hazard to use moderate drinking for diabetes patients. In this study, the islet area was enlarged by 5% ethanol treatment, but this phenomenon was not observed after OA and OA+EtOH treatment.

Although both EtOH and OA+EtOH groups promoted the increase in total islet cell number (Fig. 3C), the mechanisms between them were different. 5% ethanol mainly increased the α-cell number and OA+EtOH most likely increased both α-cell and β-cells numbers (Figs. 4B, 4C and 4E). Islet α-cell secreted glucagon increases hepatic glucose output by stimulating gluconeogenesis and glycogenolysis (Briant et al., 2016). The increase of α-cell number in the EtOH and OA+EtOH groups (Fig. 4E) promoted the glucagon secretion which leads to downregulation of liver glycogen level (Fig. 2B). Interestingly, the raised α-cell number in the EtOH and OA+EtOH groups did not result in FBG level elevation, instead, they suppressed the FBG level of STZ induced diabetes. Over half a century ago, it was observed that the glucagon receptor is located on both pancreatic α and β cells, so that glucagon can promote insulin secretion (Samols, Marri & Marks, 1965). It is highly possible that EtOH and OA+EtOH not only promote glucagon expression but also promote glucagon receptor expression in β-cells. Therefore, it is not surprising that EtOH and OA +EtOH significantly promoted the insulin RNA expression of diabetic rats (Fig. 6A).

Paracrine interactions between pancreatic islet cells have been proposed as a mechanism to regulate hormone secretion and glucose homeostasis. The reduction of GLP-1R expression was commonly found in diabetic conditions, and the decreased expression of GLP-1R is associated with the pathogenesis or progression of β-cell failure (Kubo et al., 2016). Due to the anti-hyperglycemia effect of GLP-1R, several unimolecular GLP-1R agonists are currently tested in phases 1 and 2 clinical studies with promising outcomes (Sánchez-Garrido et al., 2017). After STZ injection, loss of β-cell mass was observed synchronously with GLP-1R positive cell reduction (Fig. 5) which in turn led to a high FBG level. After OA, EtOH, or OA+EtOH treatment, the GLP-1R expression was promoted and suppressed the glucose level (Fig. 5). In normal rats GLP-1R is only expressed in β-cells, although GLP-1R positive staining was found in α-cells under diabetic conditions (Nakashima et al., 2018). Islet β-cell expressed GLP-1R which can be activated by glucagon, potentiating the secretion of insulin (Capozzi et al., 2019). OA augmented the GLP-1R positive cell number, and at the same time upregulated the insulin positive cell ratio, implicating that GLP-1 can not only promote insulin synthesis but also stimulates β-cell proliferation and inhibits β-cell apoptosis (Campbell & Drucker, 2013). In this investigation, the increase in islet cell number caused by EtOH was mostly attributable to an augmentation of α-cell mass, which associated with an enlargement in GLP-1R positive cell number. Another research also supported our finding, when the glucose level is high, GLP-1R in α-cell exert the function of inhibiting glucagon secretion (Zhang et al., 2019) which can prevent the increase of glucose level. On the other side, glucagon stimulates insulin secretion through both glucagon receptor and GLP-1R, the combined activity of glucagon and GLP-1R is essential for β-cell secretory responses (Svendsen et al., 2018) which maintains glucose homeostasis. EtOH+OA group shows a higher β-cell ratio than EtOH (Figs. 3D and 4D), and a greater α-cell ratio than OA (Figs. 4F and 4G), at the same time, the GLP-1R positive cell number is significantly bigger than the hyperglycemia group (Figs. 5B and 5D). Moreover, GLP-1 has been proposed to play an important role in α-cell trans-differentiation to new β-cells (Lee et al., 2018), the β-cell number that was increased in OA+EtOH group might likely be transformed from α-cell by the elevated expression of GLP-1R (Fig. 5B). The comprehensive mechanism of how OA and moderate drinking modulate blood glucose levels when lacking of β-cell mass was depicted in Fig. 7.

Figure 7 The mechanism of OA and moderate drinking on β-cell mass deficiency induced hyperglycemia.

Conclusions

This study demonstrated that GLP-1R level is crucial for the modulation of blood glucose levels. The glucose lowering effects of OA and moderate drinking targeting on the pancreatic GLP-1R expression. The anti-diabetic effect of OA and 5% ethanol mixture discovered in this research offers possibilities to develop functional low concentration alcoholic drinks for diabetes patients.

Supplemental Information

Supplemental Information 1 ARRIVE 2.0 Checklist

Click here for additional data file.

Supplemental Information 2 Raw data for Figure 1

Click here for additional data file.

Supplemental Information 3 Glycogen level and control for Figure 2A-B

Click here for additional data file.

Supplemental Information 4 Glycogen picture of EtOH and OA+E for Figure 2A

Click here for additional data file.

Supplemental Information 5 Glycogen picture of OA and hyperglycemia for Figure 2A

Click here for additional data file.

Supplemental Information 6 Insulin level, control and hyperglycemia IHC for Figure 3A-E

Click here for additional data file.

Supplemental Information 7 Insulin OA, EtOH and OA+EtOH IHC for Figure 3A

Click here for additional data file.

Supplemental Information 8 Raw data for Figure 4

Click here for additional data file.

Supplemental Information 9 GLP1R level and control IHC for Figure 5A-E

Click here for additional data file.

Supplemental Information 10 GLP1R EtOH and OA+E IHC for Figure 5A

Click here for additional data file.

Supplemental Information 11 GLP1R OA and hyperglycemia IHC for Figure 5A

Click here for additional data file.

Supplemental Information 12 Raw data for Figure 6

Click here for additional data file.

Supplemental Information 13 Raw data for Figure 7

Click here for additional data file.

The authors thank all the coworkers in the medical experiment center for their support with materials and reagents.

Additional Information and Declarations

Competing Interests

Author Contributions

Animal Ethics

Data Availability

The authors declare there are no competing interests.

Li Xu conceived and designed the experiments, performed the experiments, analyzed the data, prepared figures and/or tables, authored or reviewed drafts of the article, and approved the final draft.

Ruibin Hu performed the experiments, analyzed the data, prepared figures and/or tables, and approved the final draft.

Shreyas Venkataraman Jois analyzed the data, authored or reviewed drafts of the article, and approved the final draft.

Lei Zhang conceived and designed the experiments, analyzed the data, authored or reviewed drafts of the article, and approved the final draft.

The following information was supplied relating to ethical approvals (i.e., approving body and any reference numbers):

The Experimental Animal Ethics Committee of Shaanxi University of Chinese Medicine approved the study (SUCMDL20160113001).

The following information was supplied regarding data availability:

The raw measurements are available in the Supplemental Files.

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
