# Peer review of "Oleanolic acid and moderate drinking increase the pancreatic GLP-1R expression of the β-cell mass deficiency induced hyperglycemia"

_PeerJ, doi:10.7717/peerj.15705_

## Round 0.1 · original submission · Major Revisions

Dear Xu & Zhang,

Thank you for submitting your manuscript to PeerJ. The referees have reviewed your manuscript carefully and recommended some modifications on the manuscript before further processing. Hence, the decision “Major revision” was taken for your submitted manuscript.

The referee would like to see easily the modifications made to your manuscript in the revised version. Therefore, I invite you to respond to the referee(s)' comments and revise your manuscript carefully.

Do not forget to highlight ALL the changes you make, using track changes.

Please provide also an answer/report to the referee(s)’ comments, which summarizes the changes you have made IN the manuscript itself. The answer/report to the referee(s) may also include any other response that you want the editor and the reviewer(s) to note. You should submit the answer/report to the referee(s)’ comments as a separate document.

Thank you for submitting your manuscript to PeerJ and giving us the opportunity to consider your work.

We look forward to receiving your revision.

Sincerely,

Reviewer 1 ·

Basic reporting

no comment

Experimental design

no comment

Validity of the findings

no comment

Additional comments

In this manuscript, Xu et al studied the protective effect of OA and moderate drinking in STZ induced diabetes model. Although both OA and moderate drinking have been tested in a similar setting, their combinatorial effect was unknown. The authors found OA and moderate drinking combo displayed a synergistic effect on several aspects of pancreas function, e.g., glycogen storage and GLP1R expression, as well as insulin synthesis. There were also some aspects that only responded to one regimen but not the other. And through correlation analysis, they concluded that GLP-1R is critical for the observed impact, and it might be feasible to develop low-concentration alcoholic drinks for diabetes patients. The manuscript is well-written, and the description of the experiments is clear, however, there are several points I list point by point I think would help improve the manuscript.

Major points:
1. The authors wrote in the manuscript “OA and EtOH alone or combination attenuated the STZ induced hyperglycemia”, and “OA showed the ability to suppress FBG level that was caused by STZ” to me, these are very modest if any, could the authors state these conclusions more accurately.

2. The authors should explain the dosage used for “moderate drinking” in their manuscript. A citation or some explanations would help.

3. Can the authors elaborate more on their envisioned working model? Which aspects of pancreas function showed synergistic, which did not, and what other tissues might be involved? I think they mentioned the liver in their discussion. A summary figure incorporating these would help.

Minor points:

1. In the abstract, “the synergistic effects of OA and moderate drinking on islet -cell deficiency induced diabetes are not clear”, perhaps rephrase this to not reveal the key conclusion at the start.

2. Fig.1A Can the authors provide specific p values or statistics for these comparisons to show although it's not significant there is a downward trend? The same applies to Fig.1B.

3. Fig.1B Can the authors help me with the label of the y-axis? Is it after 6 weeks or between 1-6 weeks? I think it’s after 6 weeks, was there a typo?

4. Fig.1E Did the authors perform any quantification analysis like fig.3e, if yes, should include in this figure? Otherwise, these should label as “representative”

5. Fig.2 It’s not clear to me from the label the three last groups (OA, ethanol, OA+ethanol) are on top of STZ-induced hyperglycemic condition. Please add this information. The same applies to other figures.

6. Line 162, Can the authors say more about why they chose to determine glycogen levels in the animal, which would help with the flow of the manuscript?

7. Line 227, INS1 level, which figure is the author referring to, fig5A or fig6a?

Reviewer 2 ·

Basic reporting

There is a lot of overlap in Figure 1, especially in Figure 1C. Which points (in each week) correspond to which group is not

Experimental design

What do you mean by ‘Male SD rats’? It might refer to Sprague Dawley rats but atleast at the first instance of abbreviations, these should be expanded.

“50 rats were injected with 55mg/kg STZ to generate hyperglycemia according to our previous method (Xu et al. 2022). Hyperglycemia rats were randomly divided into the STZ group (10 rats), OA group (10 rats), EtOH group (8 rats), and OA+EtOH group (10 rats). 10 rats were injected with an equal dose of 0.1M sodium citrate buffer (pH=4.4) as the control group. “
How were 50 rats divided into 10,10,8 and 10? Were only those rats taken in whom hyperglycaemia was successfully achieved? Or, was there some other method of choosing rats, or is there loss of data? Things like these should be clarified.

“Control and STZ groups were treated with saline”
How much saline was injected? The volume is an important consideration in in vivo experiments.

“Rats were anesthetized with 2% isoflurane after 6 weeks of treatment, then blood, livers, and pancreas were collected. After sample collection, all animals were euthanized by 5% of isoflurane.”
How were the organs harvested? Results depend on this too. These should be explained properly sequentially.

In methods, the authors have not discussed whether they are checking the assumptions (like normality of distribution of data) for the statistical tests they have decided to use

Validity of the findings

.

Additional comments

Major language improvements are needed

---

## Round 0.2 · accepted · Accept

Dear Authors,

Thank you for revising the manuscript. I am pleased to inform you that the authors have diligently addressed the majority of the comments raised by the reviewers during the evaluation process. They have carefully considered each suggestion, critique, and recommendation, and have made substantial revisions to the manuscript accordingly. With the revisions implemented, the paper now may present a more robust and refined contribution to the field. Therefore, I recommend considering this revised version for publication, as it adequately addresses the concerns raised by the reviewers and enhances the overall strength of the paper.

Reviewer 1 ·

Basic reporting

NA

Experimental design

NA

Validity of the findings

NA

Additional comments

I have previously reviewed this manuscript. The authors addressed most of my major concerns and the manuscript is now suitable for publication. I felt which would be more helpful for me as a reviewer is to provide a rebuttal letter.